# Antireflective vertical-cavity surface-emitting laser for LiDAR

Cheng Zhang [1], Huijie Li [1] & Dong Liang [1]✉

Multijunction vertical-cavity surface-emitting lasers (VCSELs) have gained popularity in automotive LiDARs, yet achieving a divergence of less than 16° (D86) is difficult for conventional extended cavity designs due to multiple-longitudinal-mode lasing. Our innovation, the antireflective vertical-cavity surface-emitting laser (AR-VCSEL), addresses this challenge by introducing an antireflective light reservoir, where the electric field intensity is substantially higher than the gain region. This reduces the required cavity length for minimal divergence, preserving the single-longitudinal-mode lasing. A 6-junction AR-VCSEL array showcases a halved divergence and tripled brightness compared to its conventional counterpart. Various multijunction AR-VCSEL array designs achieve a divergence range of 8° to 16° (D86). Notably, a 7 μm AR-VCSEL emitter achieves 28.4 mW in single transverse mode lasing. AR-VCSEL stands out among semiconductor lasers, offering a well-balanced power density and brightness, making it a cost-effective solution for long-distance LiDARs. The antireflective cavity concept may inspire diverse applications in photonic devices beyond LiDARs.

Compactness, fast response, and high energy conversion efficiency have made vertical-cavity surface-emitting lasers (VCSELs) a major light source for high-speed data communication[1–3] and sensing[4–6] over the past three decades. Particularly in the last few years, driven by applications such as face identification (face ID), infrared illumination, time of flight (ToF) approximation, and 3D sensing for smartphones, tablets, robotics, etc., unprecedented billion-unit VCSEL[7] chips and modules have been produced and assembled. As the supply chain for mass production matures, the massive application of VCSELs is expanding to autonomous driving[8], computing[9], virtual and augmented reality[10,11], industrial fast heating[12], esthetic medicine[13], etc. Among them, LiDAR systems equipped with VCSEL array solid-state light sources are commercialized in autonomous-driving vehicles[14,15].

There are numerous performance metrics for high-resolution LiDAR light sources, such as power, power density, divergence angle, beam quality, beam parameter product, spectral width, brightness, spectral brightness, wavelength, temperature stability, pulse width, energy conversion efficiency, on-off speed, module size, and power per active area. Among them, brightness, defined in Eq. (1), as the power flow per unit emission area $A$ per unit solid-angle $\Delta\Omega$

(steradian, sr)[16,17], consolidates the first five metrics.

$$\text{Brightness} = \frac{\text{Power } density}{\Delta\Omega} = \frac{\text{Power}}{A\Delta\Omega} \quad (1)$$

And spectral brightness, defined by

$$\text{Spectral } brightness = \frac{\text{Brightness}}{\Delta\lambda} = \frac{\text{Power}}{A\Delta\Omega\Delta\lambda} \quad (2)$$

consolidate the first six metrics.

Micro-scanning, 1D/2D addressable, and flash LiDAR are examples of direct-time-of-flight (dToF) LiDAR technologies that use semiconductor lasers. While they all require high power and low cost, they do have varying demands on the various characteristics of the light source. Brightness is especially critical for the scanning LiDAR systems that utilize collimated laser beams. The higher the brightness is, the farther and the higher the angular resolution a scanning LiDAR system can possibly perceive if the imaging sensor has sufficient resolution. On the receiving side, narrow bandpass filters are typically applied in

¹Vertilite Co. Ltd., Wujin District, Changzhou, Jiangsu, China. ✉e-mail: ld@vertilite.com

front of the detector for a high signal-to-noise ratio, enabling longer perceiving distance. The system signal-to-noise ratio can be improved proportionally to the reduction of the filter bandwidth. Conventionally, the bandwidth of the filter covers the entire range of the laser source wavelength shift over −40 °C to 125 °C required by the Automotive Electronics Council standard. However, if the temperature coefficient of the wavelength shift of the filter can closely match the laser source, the filter bandwidth can be significantly narrower. Further, if the filter wavelength-angle shift can be minimized, the filter bandwidth can even approach the laser spectral width itself. In this case, spectral brightness becomes more important to evaluate a laser source's ultimate capability to enable a high signal-to-noise ratio.

For an array-type laser source, $A$ is defined as the area of the smallest circle that encloses all emitters. The solid angle can be expressed as $\Delta\Omega = \iint \sin\theta d\theta d\varphi \approx \pi\theta^2$, where $\theta$ is the divergence half-angle, with the approximation that $\sin\theta \approx \theta$ when $\theta$ is small. The denominator $A\Delta\Omega$ (etendue) in Eq. (1) is preserved as the light beam travels through any ideal collimating lens systems[18]. It can only increase if any optics are nonideal during propagation. Similarly, the power can only be preserved or deteriorated for any optical loss during propagation. As a result, the brightness and the spectral brightness can only deteriorate or, at best, be preserved in a lossless ideal lens system. Therefore, in addition to perfecting the optics and increasing the laser emission power, reducing the etendue of the original laser beam from the bare chip is critical to achieving high brightness and high spectral brightness for distant objects.

Regarding the choice of the most suitable semiconductor laser source for commercial LiDAR[19,20], there is an ongoing competition between edge-emitting lasers (EELs) and VCSELs. EELs were introduced to LiDAR applications earlier than VCSELs because single-bar EELs usually produce substantially higher power than single-emitter VCSELs. However, recently, the preference has started to shift to VCSELs. Compared to high-power Fabry–Pérot (FP) EELs, VCSELs have a narrower spectral width (<2 nm) and better wavelength stability with temperature (0.06–0.07 nm/°C). Over a wide temperature range of −40 to 125 °C, the VCSEL wavelength drifts by only 12 nm, allowing a narrow bandpass filter at the receiver[21] even without a wavelength-shift-matching. Though typical filters have a smaller temperature coefficient of wavelength shift[22], a LiDAR filter specially designed to match VCSELs' 0.06–0.07 nm/°C is possible. However, to make it as large as FP EELs' 0.2–0.3 nm/°C is pragmatically difficult. EELs with improved wavelength stability require carefully designed Bragg gratings and additional fabrication processes, such as e-beam lithography and regrowth[23]. In addition, VCSELs can produce superior light beams with circular symmetry, whereas the EEL beam profile is elliptic. Finally, the inherent two-dimensional array manufacturability endows VCSELs with an unmatchable advantage over EELs in two-dimensional (2D) point cloud generation and chip-scale optical integration without complicated packaging. Because of these advantages, many LiDAR manufacturers are adopting VCSEL arrays as their light sources[24,25]. However, compared to EELs, conventional VCSEL array designs still need higher power density, smaller divergence angles, and higher spectral brightness to prevail in the competition for LiDAR light sources.

There have been several ways to improve the VCSEL power density to the order of $10^3$ W/mm², or even close to high-power EEL arrays[26]. A natural benefit arises from LiDARs using ToF signaling that operates at nanosecond short pulses and a low duty cycle, resulting in a relatively low time-averaged power that avoids overheating of laser chips. The game changer for VCSELs is the tandem or multijunction structure. By connecting several P-N junctions in series with tunnel junctions (TJs) vertically without increasing the emission area, a magnified slope efficiency (SE, in units of W/A) proportional to the number of P−N junctions can be produced, leading to a significantly higher power density at a moderate current injection level. From an energy-

saving perspective, for a fixed desired optical power output, a tandem structure effectively reduces the input current by elevating the input voltage (not an issue for automobiles, although more limited for consumer electronics). The lower current generates less waste heat from the parasitic series resistance in the driver circuit, increasing the systematic energy conversion efficiency. Leading companies in the industry are now mass-producing 5−7-junction VCSELs[15,27,28], and larger numbers of junctions are under development. In addition to multijunction, the VCSEL power density can be further boosted by increasing the effective emission area over the entire VCSEL array area (filling factor) and stretching the operating current as long as the device lifetime allows.

Although multijunction structures can also be adapted for EELs, their power density is limited by the necessary separation of optical modes from each junction because TJs must be placed at optical field minima between optical modes to avoid unacceptable absorption loss[29]. Without the standing wave that provides the close-packed hills and valleys for quantum wells and TJs to reside in VCSELs, respectively, junctions in an EEL must be spaced by as much as a few micrometers. As a result, multijunction in an EEL appears as separate emitters in near-field imaging and does not increase its power density (though the power per active area does). The number of junctions in EEL is also more limited by total thickness compared to VCSELs.

As the power density of VCSELs has become sufficient for LiDARs, there is an urgent need to reduce their beam divergence for higher brightness. Because a lower beam divergence usually comes with fewer high-order modes and a narrower spectral width[30], its impact is doubled on the spectral brightness. The typical full divergence angle ($\Theta = 2\theta$) in D86 (defined as the angle at which the D86 beam width in the far field proportionally increases with the distance from the light source, where the D86 beam width in the far field is defined as the diameter of the circle that is centered at the centroid of the beam far field profile and contains 86% of the beam power) of oxide-based VCSELs is 20°–30°, which is quite large for most long- and medium-range (>100 m) scanning LiDARs. Multijunction VCSELs that have multiple oxide layers for current confinement may suffer from even larger divergence angles than single junction VCSELs due to stronger transverse optical confinement.

Reducing the number of oxide layers in multijunction VCSELs may cause power conversion efficiency (PCE) loss[31]; otherwise, additional electrical confinement, such as ion implantation, is needed. A conventional method to reduce the VCSEL beam divergence angle while maintaining the efficiency is to extend the cavity length[32] so that the contrast of the effective refractive index ($\Delta n$) between the inside and outside of the optical aperture of the VCSEL is reduced. Such reduced index confinement suppresses the generation of high-order transverse modes. Therefore, the extended cavity effectively acts as a higher-order mode suppressor or a "low pass" mode filter. After filtering out the higher-order modes that exhibit lower beam quality (a larger $M^2$ factor and a larger divergence angle)[33], the lower-order mode beams with a smaller divergence angle will dominate the laser modal operation. A small divergence angle of 12° for a single-junction VCSEL array was reported in 2019 with a higher-order mode filter[4]. Other methods to reduce the divergence angle include the usage of a high-contrast grating (HCG)[34], the usage of a slow light optical amplifier[35], microlens integration[36], or the utilization of different types of current confinement, such as ion implantation and buried TJs. However, most of these methods have their respective challenges, such as manufacturing complexity, high cost, low power density, and difficulty in realizing uniform light emission patterns.

Here, our work starts by applying an extended cavity length to a multijunction oxide VCSEL, aiming to mitigate its divergence; however, we encounter a challenge with multi-longitudinal mode issues. Then we propose the antireflective vertical-cavity surface-emitting laser (AR-VCSEL) and demonstrate multijunction AR-VCSEL arrays with

ultrasmall divergence working in a single longitudinal mode. Next, we conduct a comparative analysis of AR-VCSEL arrays with various junctions and optical aperture sizes, highlighting their superior performance in comparison to conventional extended cavity VCSELs. Notably, we demonstrate a single-transverse-mode multijunction AR-VCSEL emitter with a peak power of 28.4 mW. Lastly, we compare AR-VCSELs with other semiconductor lasers for LiDAR applications.

## Results

### Theory and conventional method

For fair comparison, we use a common chip form to test all epitaxial structures throughout this article: a VCSEL array that consists of 37 emitters with emitter-to-emitter distance of ~40 µm, each with an optical aperture of 22 µm, forming a hexagonally shaped emission area with a diameter of approximately 250 µm (Supplementary Fig. S2). Figure 1a shows a long-cavity 6-junction VCSEL emitter structure with bottom and top distributed Bragg reflectors (DBRs), an active region including multiple P–I–N junctions, and an extended cavity region. A simulated 2D color map of the optical intensity is shown in two imaginary cutting planes. The vertical direction electric field intensity distribution is relatively uniform throughout the active and extended cavity regions, as shown in Fig. 1c. The lasing wavelength is designed at 905 nm. The extended cavity region acts as a high-order mode filter, as it suppresses the generation of higher-order transverse modes.

The effective refractive index for VCSELs[37,38] can be approximated as the electric field intensity-averaged refractive index, as shown in Eq. (3)[39].

$$n_{\text{eff}} = \frac{\int n(z)|E(z)|^2 dz}{\int |E(z)|^2 dz} \qquad (3)$$

$n_{\text{eff}}$ is the effective refractive index, $n(z)$ is the refractive index in the $z$-axis direction, and $|E(z)|^2$ is the electric field intensity in the $z$-axis direction (the light emission direction). The integration range is over the entire VCSEL structure.

The effective refractive index difference between the inside and outside of the VCSEL light-emitting aperture is determined by Eq. (4).

$$\triangle n_{\text{eff}} = n_{1_{\text{eff}}} - n_{2_{\text{eff}}} \approx \Gamma_{\text{ox}} \times (n_1 - n_2) \qquad (4)$$

$n_{1_{\text{eff}}}$, $n_{2_{\text{eff}}}$ and $\triangle n_{\text{eff}}$ are the effective refractive indices inside and outside of the optical aperture in the horizontal plane and their difference. $n_1$ and $n_2$ are the refractive indices of the high-aluminum-content AlGaAs (such as $Al_{0.98}Ga_{0.02}As$) inside the aperture and the amorphous oxide[40] formed by the wet oxidation process outside the aperture, respectively. Since the oxide layers are located at optical node positions, the electric field distribution is only slightly affected by the oxide layers and is maintained almost unchanged by the oxidation process. Therefore, $\triangle n_{\text{eff}}$ mainly arises from the refractive index change in the high-aluminum-content AlGaAs layers through the oxidation process, as indicated by $(n_1 - n_2)$.

We define $\Gamma_{\text{ox}}$ as the oxide confinement factor, which is the energy of the optical field in the oxide layer as a percentage of the total optical energy in the whole cavity:

$$\Gamma_{\text{ox}} = \frac{\int_{\text{Oxide}}|E(z)|^2 dz}{\int_{\text{Total }cavity}|E(z)|^2 dz} \qquad (5)$$

According to step-index waveguide theory[41], linearly polarized (LP) transverse modes are present in radially symmetric index profiles with weak index guiding, which can apply to the case of oxide-confined VCSELs. The beam quality of each LP mode depends on the mode order. The lowest order mode, or the so-called fundamental mode, has

the highest beam quality, or the lowest $M^2$ factor ($M^2 = 1$), and, therefore the smallest divergence angle once the mode is coupled from the waveguide into free space. The higher the mode order, the higher the $M^2$ factor ($M^2 > 1$) and the larger the divergence angle. The number and orders of allowed LP modes depend strongly on the effective refractive index contrast between the core and cladding areas of the index profile, which correspond to the inside and outside of the light-emitting aperture of a VCSEL. By minimizing its $\Gamma_{\text{ox}}$ and $\triangle n_{\text{eff}}$, the number of allowed LP modes is reduced, resulting in a smaller divergence angle. Increasing the cavity length, placing oxide layers at the standing wave E-field nodes, and reducing the number and thickness of oxide layers can all contribute to the minimization of $\Gamma_{\text{ox}}$ and $\triangle n_{\text{eff}}$.

Nevertheless, a long cavity length poses new risks. The primary issue is that the longitudinal mode spacing or the free spectral range (FSR) decreases as the cavity length $L_{\text{eff}}$[42] increases, hindering single-longitudinal-mode operation. The emission spectrum of the VCSEL may show multiple lasing wavelengths in addition to the desired lasing mode, appearing on one or both sides of the designed lasing wavelength.

Figure 1d shows the reflectance spectrum of the entire VCSEL structure in Fig. 1a, with FP dips indicating allowed longitudinal modes. The FSR is as narrow as ~7.5 nm. A measured photoluminescence (PL) spectrum from the active region is ~20 nm wide at the half maximum (FWHM), which is larger than the mode spacing. If two longitudinal modes are covered by the emission spectrum of the active region and are within the stopbands of the top and bottom DBR mirrors, then they lase simultaneously.

Although with a small $\Gamma_{\text{ox}}$ of 0.131%, the divergence angle of such an array is as small as 18.5° (D86 full angle), as shown in Fig. 1b, multiple-longitudinal-mode lasing (Fig. 1e) is not acceptable for most applications, as it can cause potential problems such as temperature instability and efficiency loss from receiving filters.

Even though such multiwavelength lasing can be somewhat rectified by narrowing the top DBR stopband width with a reduced index contrast, the divergence is eventually limited by epitaxial thickness-induced stress, wafer bowing, and subsequent fabrication difficulties. Realizing a D86 full angle of less than 16° for a single-longitudinal-mode oxide VCSEL with more than 5 junctions is difficult. A better low-divergence design is needed to utilize the cavity length more efficiently.

### AR-VCSEL with an anti-reflective mirror and a light reservoir

Here, we propose a unique VCSEL structure with an antireflective cavity, including a multijunction active region, an antireflective mirror, and a light reservoir where the E-field intensity is exceptionally high, much higher than that in the active region. The total E-field energy stored in such an antireflective cavity is multiple times that in an ordinary extended cavity with an equal spatial volume. We will demonstrate that such an antireflective vertical-cavity surface-emitting laser (AR-VCSEL) is an ideal light source for LiDAR. The 6-junction AR-VCSEL in Fig. 2a has a cavity that consists of an active region consisting of alternating InGaAs/AlGaAs multi-quantum wells, GaAs TJs and oxidation confinement layers, an n-doped antireflective mirror with alternating high- and low-aluminum-content quarter-wavelength-thick AlGaAs layers, and a 2-µm-thick light reservoir made of AlGaAs. The detailed epitaxial structure is shown in Supplementary Fig. S1.

Immediately to the left of the active region, instead of a simple extension of the cavity, we add an antireflective mirror to extract light from the active region and store it in the light reservoir, like an optical dam that holds photons and raises their intensity level. As shown in Fig. 2c, the electric field peak intensity inside the light reservoir is approximately 3 times the intensity in the active region and approximately 4–5 times that in the extended cavity region in Fig. 1c, with both output levels normalized to unity.

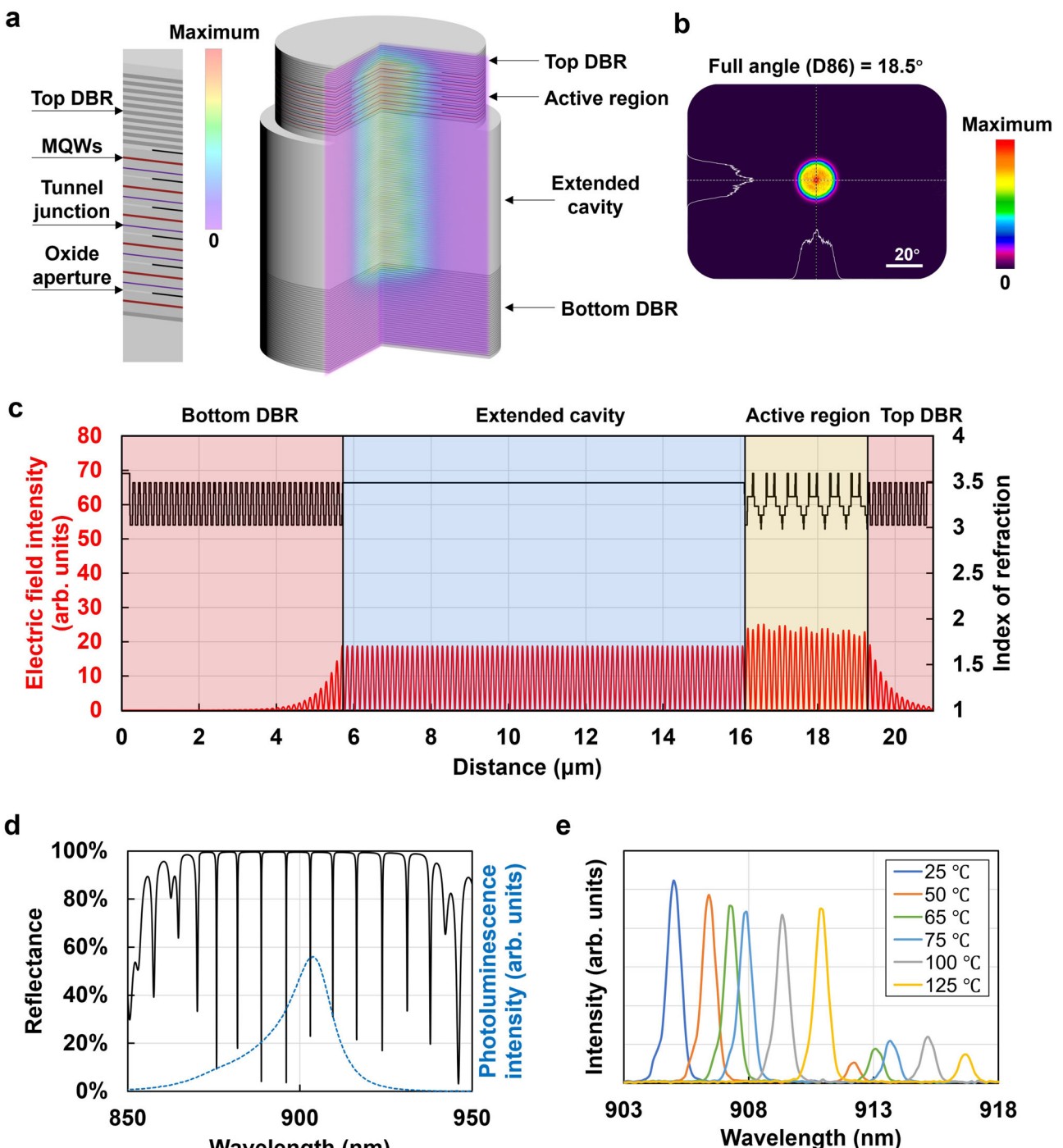

**Fig. 1 | Long-cavity 6-junction VCSEL. a** Schematic structure consisting of top and bottom-distributed Bragg reflectors, an active region, and an extended cavity region, and two-dimensional electric field intensity distribution in a sectional view. **b** Array far-field pattern 50 mm away. **c** Refractive index profile and electric field intensity distribution along the vertical axis with the output level normalized to unity (the epitaxial direction is from left to right). **d** Reflectance spectrum of the as- grown VCSEL structure (solid black line), showing FP longitudinal modes, as well as a measured photoluminescence spectrum from the active region shifted to be aligned with the center FP dip for illustration purposes (blue dashed line). **e** Measured temperature-dependent array lasing spectra from 25 °C to 125 °C, showing two longitudinal lasing modes.

The antireflective mirror consists of a few pairs of DBRs similar to the bottom DBRs but designed with a special π/2 (or quarter wavelength) phase shift. The photons generated from the active region traveling towards the bottom DBR interfere constructively at each antireflective layer and reach increasingly higher intensity until stabilization at the light reservoir. Figure 2d–e shows a close-up electric field comparison at the active region/mirror interface between a traditional VCSEL structure in 2d and an AR-VCSEL in 2e. With a quarter wavelength spacer layer located between the active region and the antireflective mirror, the electric field antinode positions shift from their original index-decreasing interfaces in 2d to index-increasing interfaces in 2e along the direction from the active region to the bottom mirror. Supplementary Fig. S3 further illustrates how the electric field is established in an AR-VCSEL.

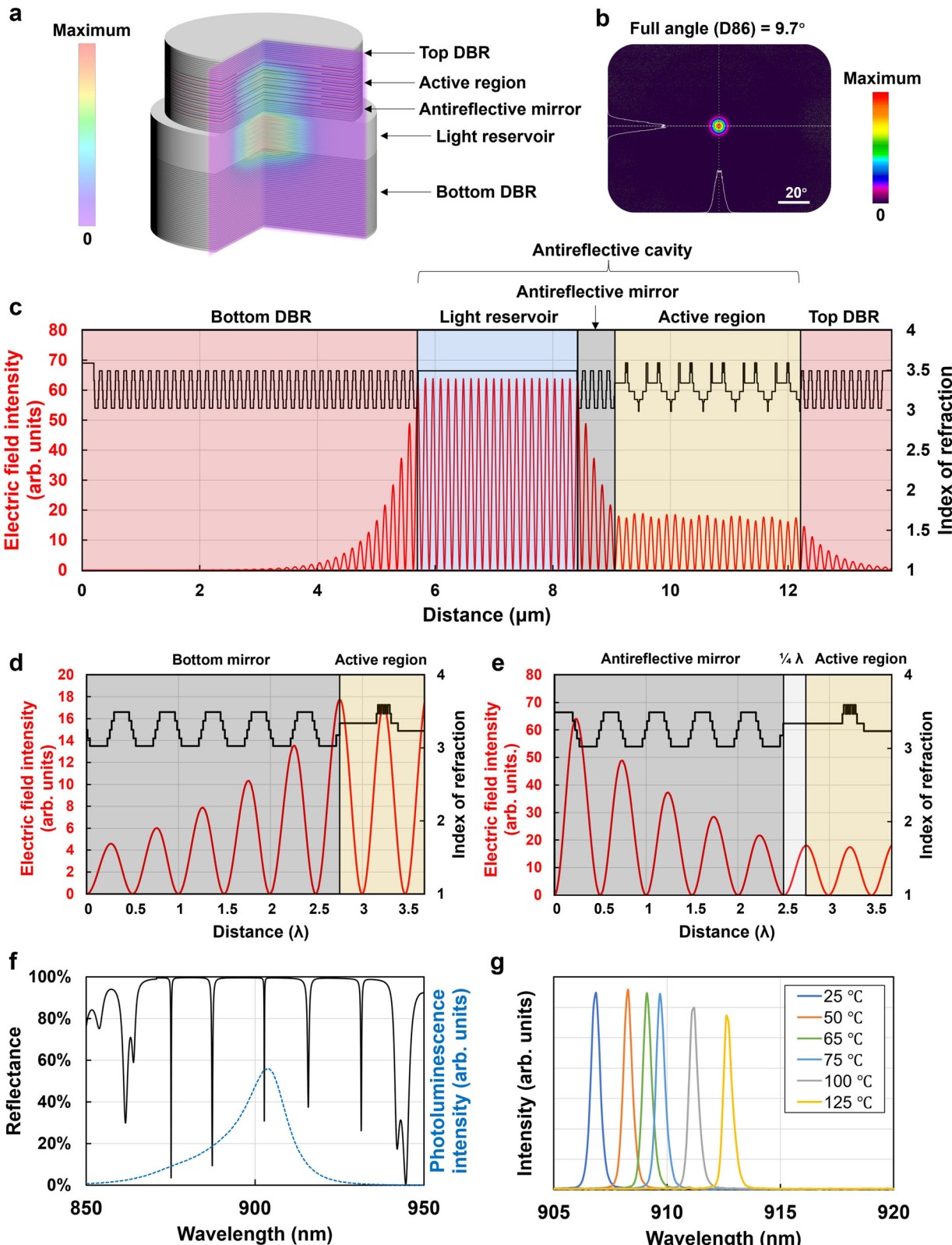

**Fig. 2 | 6-junction AR-VCSEL. a** Schematic AR-VCSEL structure consisting of top and bottom DBRs, an active region, an antireflective mirror, and a light reservoir, and two-dimensional electric field intensity distribution. **b** Array far-field pattern 50 mm away. **c** Refractive index profile and electric field intensity distribution with the output level normalized to unity (the epitaxial direction is from left to right). **d** A standard VCSEL structure with a bottom mirror next to the active region.

**e** An AR-VCSEL structure described in **a–c** with a $\pi/2$ (or quarter wavelength) phase shift between the antireflective mirror and the active region. **f** Reflectance spectrum of the as-grown AR-VCSEL structure showing several FP longitudinal modes within the stopband (solid black line), as well as a measured PL spectrum aligned with the center FP dip (blue dashed line). **g** Measured temperature-dependent lasing spectra from 25 °C to 125 °C, showing single-longitudinal-mode lasing.

Our unique design transforms a long cavity extension into a shorter extension but with a much stronger electric field. Such an antireflective cavity is more efficient in storing photons in each unit of cavity length, thus more effectively lowering $\Gamma_{ox}$. The stored photons inside the reservoir feel almost no lateral confinement. They are essentially 'free' laterally, significantly reducing the overall divergence angle.

The strong electric field intensity inside the light reservoir lessens the dependence of the divergence angle on the thickness of the extended cavity. As a result, only a moderate thickness of the light reservoir is needed while maintaining a large FSR of ~16 nm (Fig. 2f) to reach an even smaller divergence angle (Fig. 2b) with a reduced $\Gamma_{ox}$ of 0.027% while maintaining a single longitudinal mode (Fig. 2g).

Figure 3a–g shows the comparison between the AR-VCSEL and the extended cavity VCSEL on the averaged $M^2$ factor of all individual emitters within each array, the near field, and the far field 115 cm away. The $M^2$ factor, which is proportional to the FF angle as $M^2 = \pi r \theta / \lambda = \pi r \Theta / (2\lambda)$ where $\Theta$ is the full angle in D86, is stable at different current injection levels for both the AR-VCSEL and the extended cavity VCSEL (Fig. 3a). The sparser speckles in AR-VCSEL near field image (Fig. 3f) indicate fewer number of transverse modes, consistent with its reduced $\Gamma_{ox}$. In Fig. 3h–j, we compare the performance of the AR-VCSEL, our extended cavity VCSEL, the-state-of-the-art commercial multijunction VCSEL[43], and the-state-of-the-art commercial multijunction EEL for LiDAR[44], in terms of the light output power within a 10° field of view (Fig. 3h), the brightness (Fig. 3i), and the spectral brightness (Fig. 3j). The brightness and spectral brightness are calculated using Eqs. (1, 2) and $\Delta\Omega = \pi\Theta^2/4$, where $\Theta$ is the full angle in D86. Our extended cavity VCSEL has similar performances with the-state-of-the-art commercial multijunction VCSEL in all three metrics. In contrast, our AR-VCSEL has dramatically improved the performance in each of the three metrics. With identical array size and layout, our AR-VCSEL has more than double the power of our extended cavity VCSEL contained within a 10° field of view (FOV). The brightness increases by threefold from 12.5 kW mm$^{-2}$ sr$^{-1}$ to 38.5 kW mm$^{-2}$ sr$^{-1}$, and the spectral brightness increases by more than sixfold from 12.2 kW nm$^{-1}$ mm$^{-2}$ sr$^{-1}$ to 75.6 kW nm$^{-1}$ mm$^{-2}$ sr$^{-1}$ at 10 A, though there is a slight reduction in the external quantum efficiency (Supplementary Fig. S10) due to the increased higher order mode loss. Compared with the state-of-the-art commercial multijunction EEL for LiDAR, our AR-VCSEL has more than double the power within a 10° FOV (Fig. 3h). Although the brightness of our current AR-VCSEL is still lower than the state-of-the-art EEL, the spectral brightness (Fig. 3j) is more than doubled, which is believed to substantially improve the performance of long- and medium-range LiDARs equipped with narrow bandwidth filters. Despite the fact that the EEL can reach a similar spectral brightness at a higher current, it is less valuable due to the lack of narrow bandwidth filters matching its large temperature coefficient of wavelength shift.

Note that the antireflective mirror and the light reservoir are not necessarily separable. An antireflective mirror can be built into a light reservoir. The structure and phase can be flexible as long as the E-field intensity is enhanced compared to the active region. An example is shown in Supplementary Fig. S4. The key to such E-field profile engineering is to guarantee that the accumulated phase shift over the whole light reservoir, or in other words, the optical distance between the active region and the bottom DBR, is a half-wave integer. Such flexibility of the light reservoir design provides great potential for engineering and tailoring the light intensity distribution in AR-VCSELs.

Note that the AR-VCSEL design is different from a double cavity design (active-passive cavities)[45], which was proposed mainly for narrow bandwidth applications. If used for low divergence purposes, the substantive drop in the electric field between the passive and active cavities not only is inefficient in lowering $\Gamma_{ox}$ but also adversely increases the total cavity length and decreases the FSR. It is also different from a passive cavity surface-emitting laser[46], in which a thin active layer is inserted into one of the quarter-wavelength high-index layers in its top DBR, allowing dielectric materials to form a passive cavity for better temperature stability or mode control. The antireflective cavity in this article is one whole cavity that combines both active and passive regions. The thickness of its multijunction active region typically well exceeds the total thickness of its top DBR. Additionally, considering the loss mechanism of the antireflective cavity, the antireflective mirror or layers inside the reservoir do not contribute to external mirror loss and thus cannot be considered a part of the top DBR.

Benefiting from its large cavity size and high E-filed intensity, an AR-VCSEL can amplify much light from a multijunction multi-quantum-well gain region, store the majority of its photons in the reservoir with low lateral index confinement, and, as a result, output high optical power with low divergence.

To cover a wide divergence angle range for different application purposes, more than fifty AR-VCSEL and extended cavity VCSEL structures with different $\Gamma_{ox}$ were designed and experimented with. These designs include 5, 6, 8, 10, and 14 junctions, fabricated into the identical 250 μm array pattern and tested under the same driving conditions of 3 ns pulses at a repetition rate of 20 kHz at room temperature. Their measured divergence against $\Gamma_{ox}$ is displayed in Fig. 4a. There is a clear correlation between the divergence angle and $\Gamma_{ox}$. Nearly all AR-VCSELs and extended VCSELs, regardless of the number of junctions, follow the same trending line. Through careful design, we have reached precise control of the divergence angle (D86) from 8° to 25°. The smallest corresponds to 4.1° full width at high maximum (FWHM). To the best of our knowledge, this is the first time that an ultrasmall divergence angle (D86) below 10° has been achieved with the single-longitudinal-mode operation for a multijunction VCSEL array by only optimizing its epitaxial structure without any type of lens, lateral grating or 2D photonic crystal structure. As shown in Fig. 4b, the AR-VCSELs consistently show lower divergence than the extended cavity designs with equal cavity lengths. It is also noteworthy that the divergence angle does not form a line shape with the effective cavity length, as $\Gamma_{ox}$ can vary for the same effective cavity length. Figure 4c shows the 6-junction array brightness and spectral width of these AR-VCSELs and extended cavity VCSELs at 10 A. Figure 4d shows the 6-junction array spectral brightness. The same junction number is used for a fair comparison between AR-VCSELs and extended cavity VCSELs. The highest brightness that we have achieved with 6 junctions in the 250 μm diameter AR-VCSEL array is ~40 kW mm$^{-2}$ sr$^{-1}$ and ~140 kW mm$^{-2}$ sr$^{-1}$ for single emitters due to fully utilized emission area. The spectral brightness we have achieved in AR-VCSEL at 10 A is ~75.6 kW nm$^{-1}$ mm$^{-2}$ sr$^{-1}$ for arrays and ~260 kW nm$^{-1}$ mm$^{-2}$ sr$^{-1}$ for single emitters, a similar level as the state-of-the-art LiDAR EEL[44], which is typically ~120 kW nm$^{-1}$ mm$^{-2}$ sr$^{-1}$ at a higher current and not useful without filter wavelength-shift-matching. As a reference, the state-of-the-art VCSELs for LiDAR[43] have a moderate spectral brightness of only ~12 kW nm$^{-1}$ mm$^{-2}$ sr$^{-1}$. Considering that the number of junctions can increase while the capacity of the light reservoir can be extensively enlarged, and the emission size of the array can be shrunk, there is great potential to further increase AR-VCSEL's spectral brightness and power by several times or even an order of magnitude. For example, we have achieved over 100 kW mm$^{-2}$sr$^{-1}$ on a 100 μm square array of 6 Junction AR-VCSELs (Supplementary Fig. S6). Overall, AR-VCSELs provide high spectral brightness, best beam quality, and great temperature stability while remaining most cost-effective for producing higher power per unit area.

In addition to varying the number of junctions and the oxide confinement factor, we investigated another critical parameter: the diameter of the optical aperture (OA). We fabricated a series of AR-VCSEL arrays with a dimension of 250 μm, each densely populated with identical emitters featuring OA diameters ranging from 7 μm to 21 μm (Supplementary Fig. S8), all with the same epitaxial structure

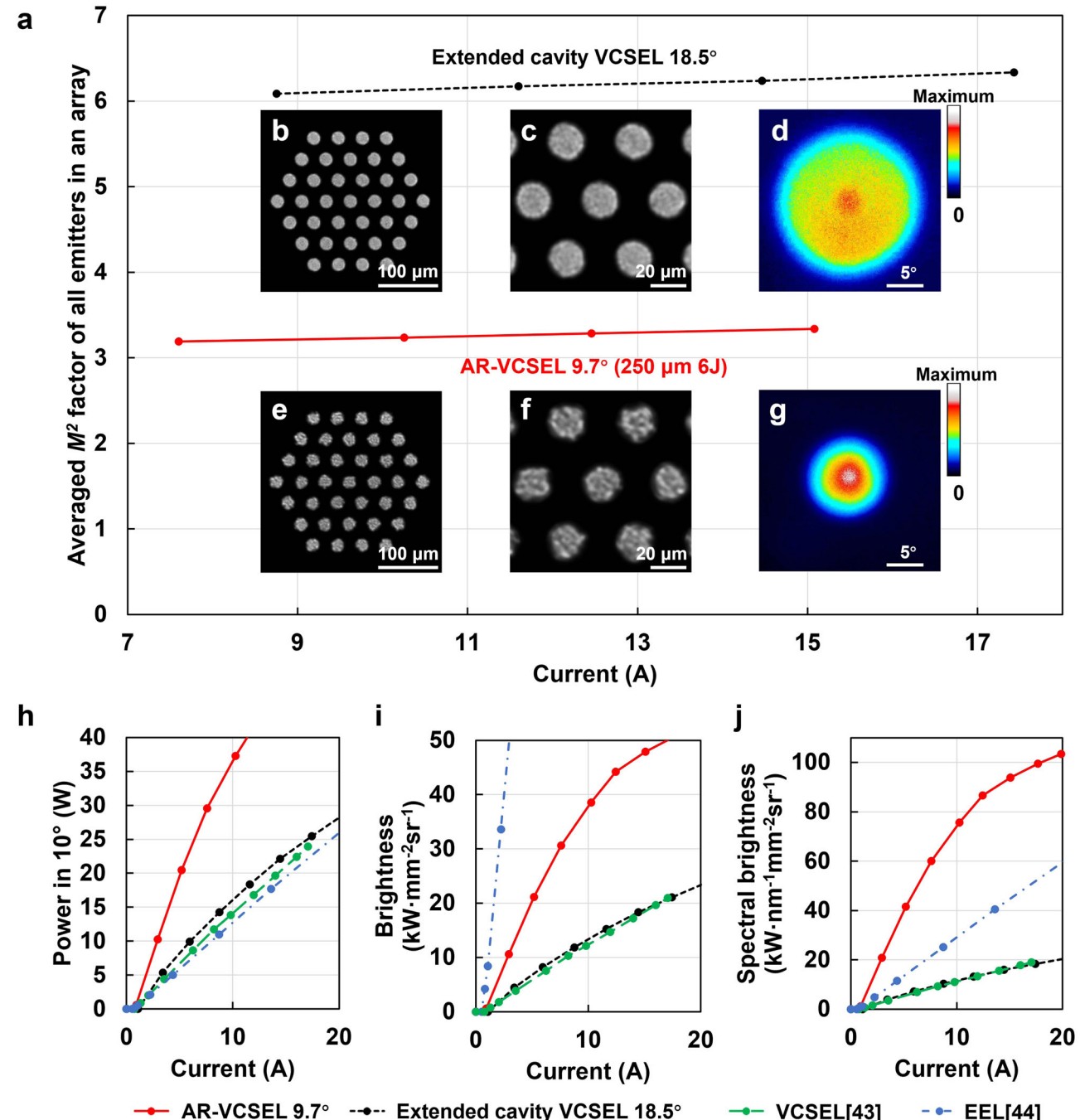

**Fig. 3 | AR-VCSEL's superior performance compared to conventional extended cavity VCSELs. a–g** Comparison between our extended cavity VCSEL array and our AR-VCSEL array on the averaged $M^2$ factor of all individual emitters within each array (**a**), the measured near field pattern of each array (**b**, **e**), the zoomed-in near field pattern (**c**, **f**), and the measured far-field pattern at 115 cm distance (**d**, **g**). The averaged $M^2$ factor of all individual emitters in an array is calculated assuming the beam waist radius ($r$) is equal to the radius of the optical aperture. The near field and far field were measured at a current of 10 A. The sparser speckles in the AR-VCSEL near-field image (**f**) indicate fewer modes. **h–j** Comparison of the measured light output power within a 10° field of view (**h**), the measured brightness (**i**), and the measured spectral brightness (**j**) of the AR-VCSEL in Fig. 2a (red solid line) with our extended cavity VCSEL in Fig. 1a (black short-dashed line), the state-of-the-art commercial multijunction VCSEL[43] (green dashed line) and the-state-of-the-art commercial multijunction EEL[44] (blue dash-dotted line) for LiDAR.

as shown in Fig. 2. Subsequently, we conducted measurements of their divergence angles and calculated the average $M^2$ values within these arrays. Our results reveal a clear correlation between OA size and $M^2$ values, as shown in Fig. 5a. As the OA size decreases, $M^2$ values also decrease. Notably, when the aperture size is reduced to 7 μm, the $M^2$ approaches a value close to 1, suggesting that the majority of emitters might operate in nearly a single transverse mode lasing regime.

To delve deeper into the behavior of the 7 μm OA array sample and confirm the possibility of single-mode lasing, we employed a 100 ns pulse driver for testing. This allowed us to gain better control at lower currents, facilitating the determination of the transition point between single-mode and multimode lasing. We utilized a free-space lens (depicted in Supplementary Fig. S9) to couple the emission of one specific emitter into an optical fiber (All other emitters in the array are optically completely blocked off by silver paste). At an optical power of

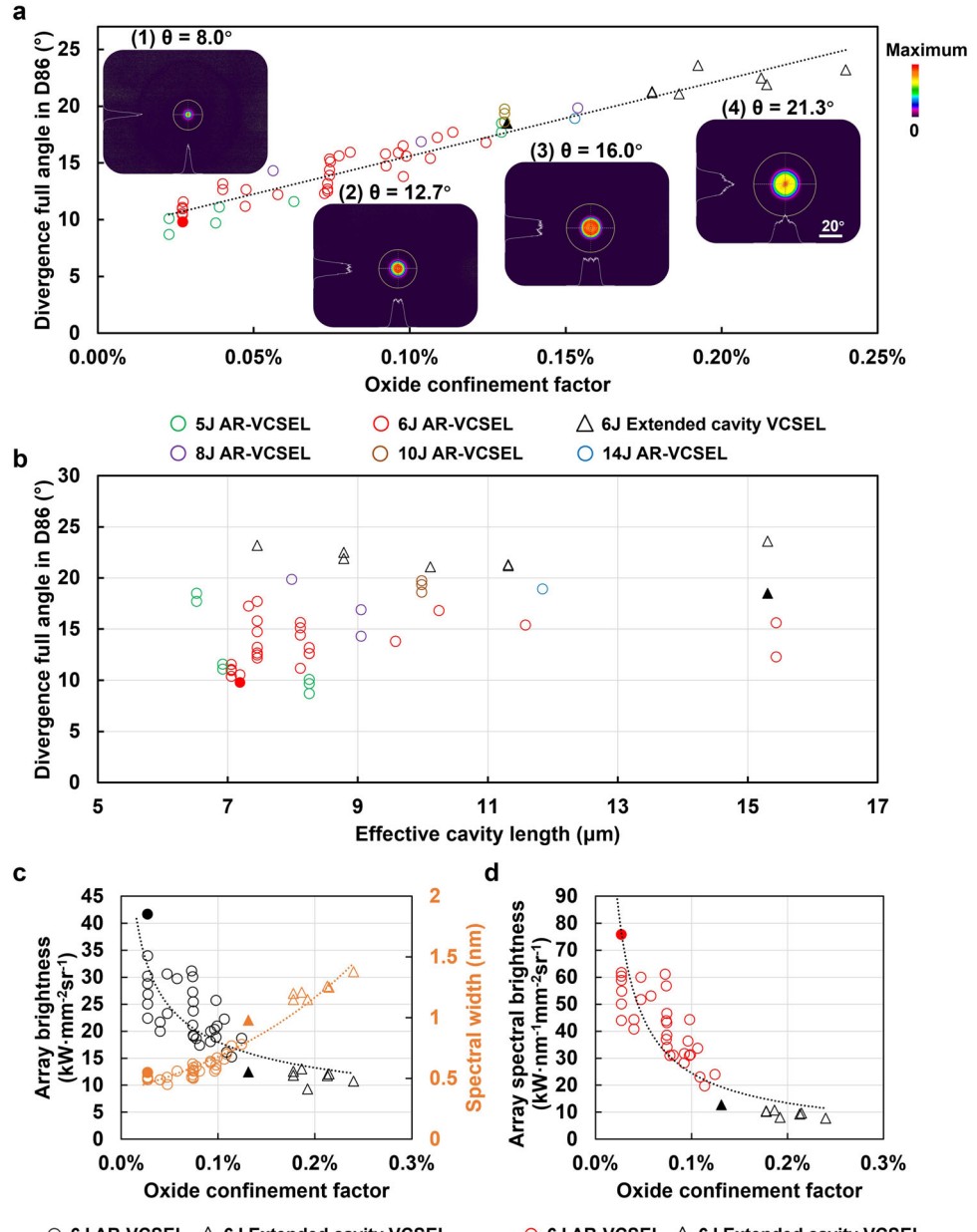

**Fig. 4 | Dependence of the divergence angle, the array brightness, the spectral width, and the array spectral brightness on the oxide confinement factor.** Circular dots represent the measured data points of 46 AR-VCSEL epitaxial designs, among which the solid dot represents the AR-VCSEL in Fig. 2. Triangular dots represent the measured data points of 8 extended cavity VCSEL designs, among which the solid represents the device in Fig. 1. **a** Dependence of the divergence angle at 10 A on the oxide confinement factor. The dotted trend line reveals a near-linear relationship between the divergence angle and the oxide confinement factor.

**b** Dependence of the divergence angle at 10 A on the effective cavity length[42]. **c** Dependence of the measured array brightness (black dots) and spectral width (orange dots) at 10 A on the oxide confinement factor. **d** Dependence of the measured array spectral brightness at 10 A on the oxide confinement factor. Green, red, purple, brown, and blue circles in (**a**, **b**) represent AR-VCSELs with 5 J (7 designs), 6 J (32 designs), 8 J (3 designs), 10 J (3 designs), and 14 J (1 design), respectively.

28.4 mW with a current of approximately 7.8 mA (calculated by dividing the total array current by the number of emitters) and a current density of around 200 A/mm², the spectrum indicates single-mode lasing with a side mode suppression ratio (SMSR) reaching nearly 40 dB (Fig. 5b). This confirms the achievement of high-power single transverse mode lasing in AR-VCSELs under the 100 ns pulse condition. The 28.4 mW peak power of this individual emitter surpassed the highest power of 14 mW for multijunction VCSEL single-mode lasing reported in the literature[47]. Notably, the 7 μm OA of the AR-VCSEL is considerably larger than the 3–4 μm typically required for traditional single-mode VCSELs without additional surface relief or complex structures.

Additionally, we obtained both near-field and far-field images of the light emitted from the aperture (Fig. 5c). The optical fields exhibit slightly elliptical shapes, likely attributed to the imperfect circular shape of the aperture. When analyzing the side views of the far-field intensity, their shapes closely resemble Gaussian curves. The far-field divergence of 9.4° for the 7 μm OA is almost at the diffraction limit, a key characteristic of single-mode operation.

It is essential to acknowledge that, at this stage, we cannot conclusively confirm the consistency of the optical mode between continuous wave (as in most previous single-mode VCSEL work) and pulsed conditions (as in this work). This aspect necessitates further

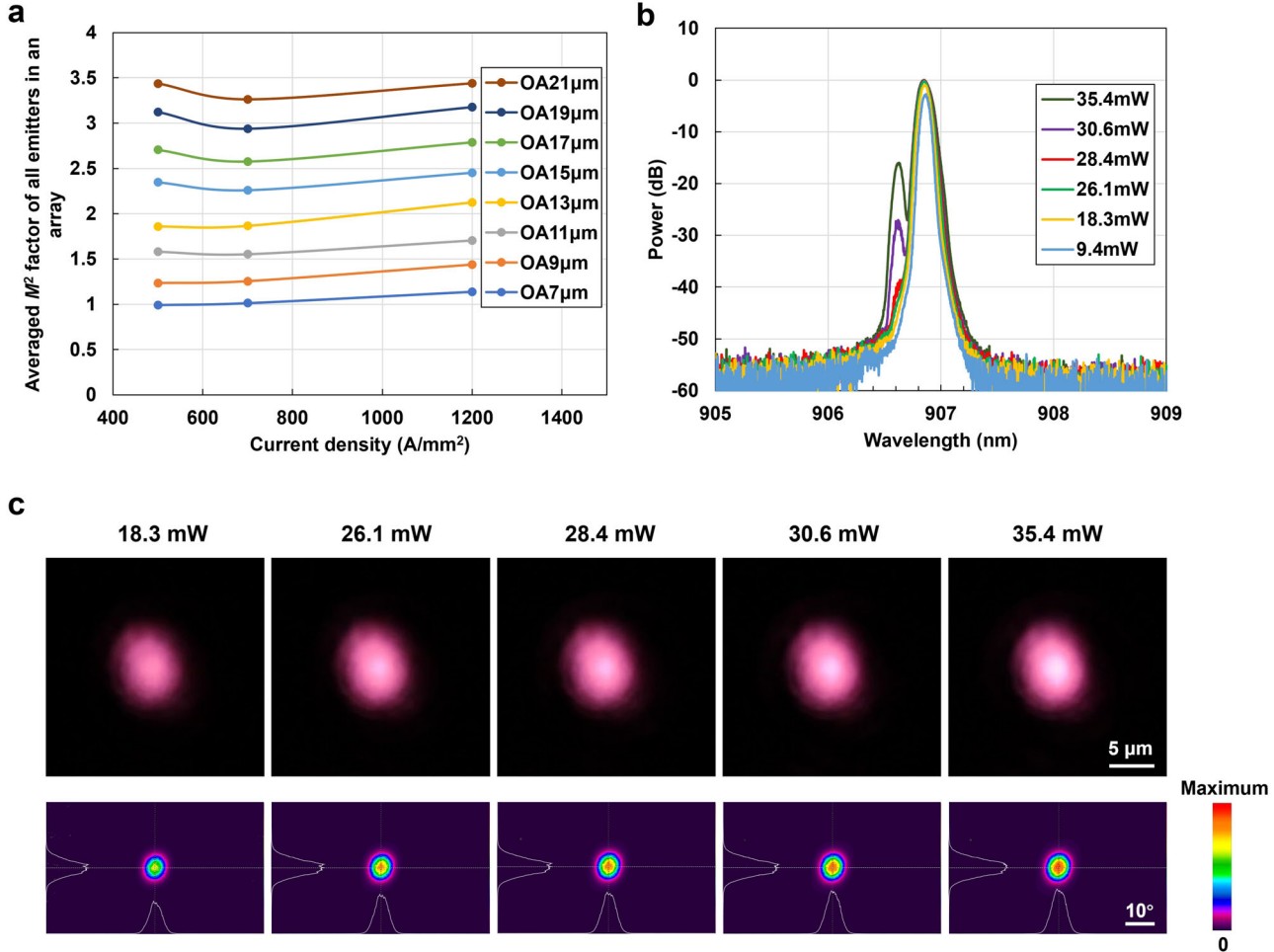

**Fig. 5 | Demonstration of single-transverse-mode AR-VCSEL. a** The averaged $M^2$ factor of all individual emitters in an array for AR-VCSELs with optical aperture (OA) sizes varying from 7 μm to 21 μm in diameter at different current densities, with the epi structure the same as in Fig. 2. **b** The measured single emitter lasing spectra of the OA 7 μm AR-VCSEL at different output powers, showing single transverse mode operation at 28.4 mW with a side mode suppression ratio of ~40 dB. **c** The measured near field and far field images of the OA 7 μm AR-VCSEL at different light output powers from 18.3 mW to 35.4 mW. **b**, **c** were measured at 100 ns pulse condition and a repetition rate of 20 kHz.

investigation in the future. Furthermore, it is important to note that in AR-VCSEL arrays, as well as in regular VCSEL arrays, the light emitted by individual emitters lacks coherence with each other. Consequently, even if each emitter operates in a single mode, the light field of the entire array cannot be considered a single mode. Instead, it represents a superposition of multiple single-mode light fields.

## Discussion
### Comparison of semiconductor lasers for LiDAR
In this section, we compare the performance of AR-VCSELs with other semiconductor lasers, particularly PCSELs, for LiDAR applications. PCSELs recently have been recognized as a potential LiDAR light source because of their extremely high brightness. However, their low power density might limit such applications. Figure 6 maps the brightness versus the power per area of various semiconductor lasers (A similar relationship between the spectral brightness and the power per area is provided in Supplementary Fig. S11). The ones in the upper-right corners will be favored in the competition for long-distance LiDARs.

*Divergence and brightness*: For a 200 m long-range scanning LiDAR, a collimated beam with divergence <0.03° is required to produce a spot size of 10 cm. PCSEL offers ~0.1° and can significantly reduce the size of the collimation lens or even possibly eliminate it for shorter distances or lower resolution. In this sense, PCSEL has advantages with a smaller etendue, allowing a smaller spot size after collimation. However, when the laser spot size becomes smaller than the sensor's spatial resolution, there are no additional benefits. For today's scanning LiDAR, a brightness of 10–20 kW mm$^{-2}$ sr$^{-1}$ is sufficient to match the sensor array with a 10 μm pitch. Our 30–60 kW mm$^{-2}$ sr$^{-1}$ brightness AR-VCSEL source is sufficient to match the next generation sensor array with ~6 μm pitch or 3 times the current pixel resolution, which may take a few years to be developed. The 100 kW mm$^{-2}$ sr$^{-1}$ brightness from our 100 μm square AR-VCSEL array (Supplementary Fig. S6) could cover the needs of sub 5 μm detector pixels. With more junctions, smaller areas, fewer oxide layers, and stronger light reservoirs, we believe AR-VCSELs with a brightness of 200–1000 kW mm$^{-2}$ sr$^{-1}$ are achievable in the near future to match higher resolution sensor arrays.

*Power density or kilowatt per chip area*: The biggest advantage of AR-VCSELs over PCSELs is the power density. The chip area determines the number of chips produced in a fixed-sized semiconductor substrate, such as the 6-in. GaAs substrate, which is widely used for VCSEL production today. If the type of substrate, epitaxial thickness and times of regrowth, layers of fabrication and complexity, and on-wafer test hours are all similar, the production cost of the whole wafers would be similar. Then, the unit cost of the chip produced is directly proportional to the chip area. Therefore, to reduce costs, a smaller chip size is preferred to produce the same amount of power required.

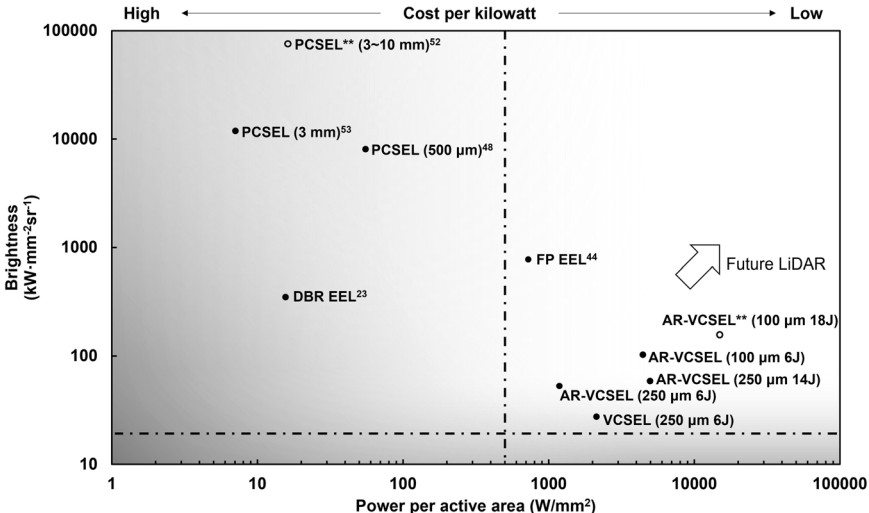

**Fig. 6 | Comparison of the maximum brightness versus the power per active area between the AR-VCSELs and other types of semiconductor lasers for LiDAR applications**[23,44,48,52,53]. ** are predicted data points and are identified as hollow dots while the solid dots are measured. The data of the state-of-the-art photonic-crystal surface-emitting laser (PCSEL) are extracted from refs. 48,52,53 for LiDAR. The EEL's power density is calculated by the power divided by the ridge area. All the surface-emitting lasers' power density is calculated by the power divided by the total emission area.

Although PCSELs may increase the power density by sacrificing beam quality, as reported experimentally, PCSEL's peak power density is only ~60 W/mm$^2$ [48]. AR-VCSEL and VCSEL can go beyond 1000 W/mm$^2$ easily (Fig. 6). As shown in Fig. 6, a 100 μm square-shaped 6-junction AR-VCSEL array we recently fabricated can provide ~45 W peak power, with a power density of ~4500 W/mm$^2$. A 250 μm diameter AR-VCSEL array with 14 junctions can produce ~240 W peak power and ~5000 W/mm$^2$ power density (Supplementary Fig. S7). Based on these numbers, we predict that a 100 μm square shaped 18-junction AR-VCSEL array can possibly reach 15,000 W/mm$^2$. Just considering the area usage of the semiconductor alone, AR-VCSEL costs 10–100 times less than PCSEL for generating the same optical power without accounting for PCSEL's expensive lithography and regrowth process. Laser and sensor chips each take about 20–40% of the total cost of a LiDAR. Under current PCSEL technology, even if we completely eliminate the cost of the transmitting-end lens (<20% of LiDAR total cost), the savings are far less than the tenfold (conservative estimate) increase in total cost brought about by the larger chip area (2–4 times that of LiDAR's total cost). In today's market, aiming toward 100 dollars in material cost for the long-distance main LiDAR requires the cost of the light source within half of it. The total power of all the light source chips in such a LiDAR is usually ~ 5 kW. Therefore, 10 dollars per kilowatt is a rough cost requirement for laser chips. AR-VCSELs, VCSELs, and even EELs in today's market can all more or less meet this cost requirement, while PCSEL needs some revolutionary technology to increase its power density by 10×−100× to be as competitive. Similar to PCSELs, the wavelength-temperature stabilized DBR lasers can also achieve decent brightness and spectral brightness. However, their power per chip area is on the far-left side in Fig. 6, making them much more expensive and less competitive for LiDARs. Complex manufacturing, such as electron-beam lithography, nanoimprint, and epitaxial regrowth, is needed for PCSEL and DBR lasers. On the other hand, AR-VCSELs can be made in high volume with existing 6-in. GaAs fabs that have been well-trained by mobile phone VCSEL productions.

In addition to LiDARs, AR-VCSELs can reduce the divergence, improve the beam quality, and minimize the crosstalk between emitters for structured light 3D sensing. For data communication applications, AR-VCSELs can potentially realize single-transverse-mode to reduce the chromatic and modal dispersion with a larger oxidation aperture than conventional single-mode VCSELs, similar to the work shown in Fig. 5, therefore increasing its power, lifetime, and transmission distance.

Furthermore, the antireflective-cavity technology can promote other cutting-edge surface-emitting laser technologies. For example, the z-direction light reservoir in AR-VCSELs can be combined with x–y plane photonic crystals[49], topological cavities[50], or metasurface structures[51] to potentially realize higher output power and efficiency for these surface-emitting lasers.

In summary, an AR-VCSEL that combines an antireflective light reservoir and a multijunction gain region has a significantly reduced divergence angle, high brightness, and high spectral brightness while maintaining the single-longitudinal-mode lasing. By solely reconstructing epitaxial layers, this unique design requires neither complex device structures nor additional fabrication steps. With a standard low-cost VCSEL process, we have realized an ultrasmall full divergence angle of 8.0° (D86) or 4.1° (FWHM), a brightness of over 40 kW mm$^{-2}$ sr$^{-1}$, and a spectral brightness of 75.6 kW nm$^{-1}$ mm$^{-2}$ sr$^{-1}$ on a 250 μm diameter 6-junction AR-VCSEL array. By applying a more compact 100 μm square array, we have experimentally improved AR-VCSEL brightness to over 100 kW mm$^{-2}$ sr$^{-1}$ and spectral brightness to over 180 kW nm$^{-1}$ mm$^{-2}$ sr$^{-1}$. By increasing the number of junctions, we achieved 5000 W/mm$^2$ power density with a 250 μm diameter 14-junction AR-VCSEL array. By varying the oxidation aperture sizes, 28.4 mW high-power single transverse mode lasing was realized in a 7 μm diameter 6-junction AR-VCSEL single emitter. To our knowledge, these are the best performances among published multijunction VCSELs. Further complemented by scalable high output power, near circular symmetrical beams, and a filter-matchable wavelength shift with temperature, AR-VCSELs exhibit more advantages over rival EELs. We also compared AR-VCSELs with PCSELs in various aspects, showing that the power density, which is the key to a low-cost LiDAR, is the most difficult challenge for PCSELs to overcome. Overall, AR-VCSELs exhibit well-balanced performance in various requirements by LiDARs. Particularly, for high-power and low-cost scanning LiDARs that require 16° (D86) of divergence or below, AR-VCSEL is the best solution available. One of our 16° AR-VCSEL products has passed the AEC-Q102 reliability tests and is now in mass production for high-performance LiDARs.

## Methods

All VCSEL/AR-VCSEL samples were epitaxially grown by an Aixtron G4 MOCVD system on 6-in. GaAs substrates in-house and fabricated following a Vertilite standard process flow at foundries.

A separate MQWs sample was prepared for photoluminescence measurement. The photoluminescence spectrum was measured by Nanometrics RPMBlue system with an OBIS LS 532 nm CW laser having an optical power of 20 mW and a spot size of 20 μm at room temperature.

All VCSEL/AR-VCSEL arrays were tested under short pulse conditions with a pulse width of 3 ns and a frequency of 20 kHz at room temperature. The driver circuit is shown in Supplementary Fig. S5. Far-field patterns were collected by an Ophir L11059 Beam Profiling Camera at an operation current of 10 A. The light output power was measured by a Newport 819D-SL-3.3 Integrating Sphere. Lasing spectra were collected by an Ocean Insight HR4Pro spectrometer at an operation current of 10 A with a resolution of 0.2 nm. Single-mode lasing spectra were collected by an Anritsu MS9740A spectrometer with a resolution of 0.07 nm. The two-dimensional electric field intensity simulation was conducted using Ansys Lumerical FDTD solutions. The full vectorial simulation was conducted using perfectly matched layer (PML) boundary conditions, and final simulation results were obtained with an auto-shutoff minimum of 1E-5 and an auto-shutoff maximum of 1E5. Simulations of both with gain and without gain in the active layer were conducted. The simulated normalized $|E(z)|^2$ field intensity profiles with gain and without gain were compared with a difference of less than 0.1%, confirming a stable static distribution.

## Data availability

The data that support the findings of this study are provided in the main text and the Supplementary Information. All the relevant data are available from the corresponding author upon request.

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

## Acknowledgements
We thank Vertilite Foundry partners for their high-quality processing services. We acknowledge the entire Vertilite team for their full support of this research.

## Author contributions
D.L. and C.Z. conceived of the idea and developed the theory. D.L. discovered a stronger E-field in an extended cavity, leading to lower divergence. C.Z. proposed an antireflective mirror to magnify this effect. D.L. conceived the light reservoir and AR-VCSEL concepts. C.Z. performed major computational modeling and simulations. H.L. performed epitaxial growth and data collection. All authors designed the experiments, analyzed data, and discussed the results. C.Z. and D.L. drafted the paper. D.L. led the project and approved the final version.

## Competing interests
The authors declare no competing interests.
