## [Peer Review File · Nature Communications]

Antireflective vertical-cavity surface-emitting laser for LiDARREVIEWER COMMENTS

Reviewer #1 (Remarks to the Author):

This is a unique paper reporting the state-of-the-art commercial product innovated by a young but leading VCSEL company. The devices are fabricated and tested using industry standards and the data quality is high. This report will benefit people working in similar fields to learn from the latest industrial perspective. I recommend it for publication in Nature Communications, after the authors address the following questions.

1) The key result of the paper is the reduction of divergence angle. However, the reason for this improvement is not clearly explained. It seems to involve oxide layers, high-order transverse modes, high-order longitudinal modes, etc.

2) It is not appropriate to invent a new name for the device, because nothing is conceptually different from VCSELs. The reservoir design of 1D grating is straightforward. Of course, incorporating it into a commercial device with good performance is nontrivial.

Reviewer #2 (Remarks to the Author):

The authors report on a new variant of a VCSEL with an additional passive cavity separated by a Bragg mirror (called anti-reflective) from the active cavity. They present a theoretical foundation and experimental results. They also give an extensive overview of the state of the art. The results achieved are impressive, in particular those of the 6 and 14 junction devices. The paper is well written and is suitable for a publication in Nature Communications, although I have my doubts whether a new name (ARCSEL) of the VCSEL is justified.

There are few issues which should be corrected before a publication.

At line 62, the authors specify the divergence angle as half angle. At line 118 they introduce the D86 beam width. In order to calculate M2 factor and brightness, the same quantity (i.e. half widths or 86% or 2nd moments) should be used for divergence and width of the beam waist. The authors must specify which quantities have been used to compute M2 factor and brightness in Figs. 3-5. By the way, I do not understand the sentence "defined as the angle at which the D86 beam width expands in the far field, where the D86 beam width is defined as the diameter of the circle that is centred at the centroid of the beam profile and contains 86% of the beam power". It is not possible to establish a connection between a point in the beam waist and an angle in the far field.

Although the authors state that the simulations were performed using Ansys Lumerical FDTD, some more details are required. Was it a semivectorial or a vectorial simulation? Which boundary conditions were used? How the static distribution is obtained?

Line 79: Give a citation for the LIDAR filters specially designed.

Line 83: Better use elliptic instead of asymmetric.

Eq. (3): E is a complex-value field because one has to deal with outgoing fields at the surfaces of the devices, so that n_{eff} is complex, too? The same question concerns Eq. (5). What is the meaning of the imaginary part of the confinement factor?

Reviewer #3 (Remarks to the Author):

In this paper, the authors propose and experimentally demonstrate a new device structure of VCSELS to realize single-longitudinal-mode lasing while decreasing their divergence angles. The authors perform systematic numerical analyses and repeated experiments. In addition, the paper is logically organized with thorough descriptions on the history of high-power VCSELS. Therefore, this reviewer believes that the paper is of interest for researchers and engineers concerning the development of high-power VCSELS for LiDAR applications. Having said that, considering the fact that various efforts have been made to increase the brightness of VCSELS in the past, this reviewer is not confident that this work will be of significance to the field of photonics and other related fields. The main concerns with this paper are described below.

1. As explained by the author, the use of multi-junction active layers or an extended cavity has already been investigated for increasing the output and brightness of VCSELS. In this paper, the authors employ antireflective layers in addition to the above-mentioned structures in order to further reduce the oxide confinement factor and to suppress higher-order modes. However, the concept of reducing the oxide confinement factor itself is the same as that of the extended cavity, and the only difference is the method to embody that concept. Furthermore, although the beam divergence angle is improved to some extent, multiple transverse modes still oscillate in the proposed device. For these reasons, the results presented in this paper can be regarded as incremental progress in this field, and they might be of little interest for researchers in other fields.

It should be also noted that the value of $M2$ shown in Fig. 3 was calculated assuming an aperture of a single VCSEL, which is actually incorrect and misleading. Considering the array device with a diameter of $250\ \mu\text{m}$, the theoretical diffraction limit of the device is around 0.3° . Therefore, the value of $M2$ should be as large as ~ 30 .

2. Relating to the above comment, in order to demonstrate complete suppression of multiple transverse modes, the authors should also carefully adjust the aperture size of each VCSEL. In order to validate the effectiveness of the proposed concept, it is necessary to demonstrate single-transverse-mode lasing in the proposed VCSELS with a larger oxidation aperture than conventional single-mode VCSELS.

3. The discussion regarding the comparison between VCSELS and PCSELS is highly biased. The authors consider that higher power density of VCSELS is one important advantage over PCSELS, but the power density is simply determined by the current density injected into the device, and is not inherent to the device structure. For example, by fabricating a PCSEL with a diameter of $100\ \mu\text{m}$ and injecting high-amplitude current into it, a power density of $1000\ \text{W}/\text{mm}^2$ or more can be easily obtained, with a much narrower divergence angle than VCSELS. In addition, higher power densities are not always desirable because it leads to shorter device lifetimes. Furthermore, as for the discussion of the cost, it is necessary to include the cost of the lens system to shape the array emission into a parallel narrow-divergence beam, because PCSELS enable lens-free operation.

Considering all these concerns, this reviewer cannot conclude that this paper has enough novelty and impact to worth publication in Nature Communications. This reviewer recommends the authors to resubmit the paper to another specialized journal such as Optics Express.

Response Letter to Reviewer Comments

(Black for comments; green for responses; blue for revisions)

Reviewer #1 (Remarks to the Author):

This is a unique paper reporting the state-of-the-art commercial product innovated by a young but leading VCSEL company. The devices are fabricated and tested using industry standards and the data quality is high. This report will benefit people working in similar fields to learn from the latest industrial perspective. I recommend it for publication in Nature Communications, after the authors address the following questions.

Response 1.0:

We appreciate the reviewer's favourable assessment of our work and willingness to suggest publication. Indeed, our devices are fabricated and tested with industrial standards with a high level of consistency and reproducibility. Also, we highly agree with the reviewer that people in similar fields will be interested in seeing VCSEL advances from the latest industrial perspective. We will address the reviewer's questions as follows:

1) The key result of the paper is the reduction of divergence angle. However, the reason for this improvement is not clearly explained. It seems to involve oxide layers, high-order transverse modes, high-order longitudinal modes, etc.

Response 1.1 :

We thank the reviewer for the comment. The mechanism of divergence angle reduction is detailed in Lines 164-183 and Lines 184-192 of the manuscript (before revision). In short, a large number of higher-order transverse modes leads to a large divergence angle, and the reduction of the divergence angle is achieved by reducing the number of higher-order transverse modes.

We first reference **the step index fibre mode theory**.

In a step-index-waveguide fibre, light is confined in a circular higher-refractive-index core region which is surrounded by an annular lower-refractive-index cladding region. Linearly polarized (LP) transverse modes are formed in the step-index waveguide. The beam quality of each LP mode depends on the mode order. The lowest order mode, or the so-called fundamental mode, has the highest beam quality, the lowest M^2 factor ($M^2 = 1$) and therefore the smallest divergence angle if the mode is coupled from the fibre into free space. The higher the mode order, the higher the M^2 factor ($M^2 > 1$) and the larger the divergence angle if it is coupled out of the fibre.

For a fixed core diameter, the allowed number and order of the LP modes in a fibre are directly related to the refractive index difference (Δn) between the core and cladding. The lower the Δn , the smaller the number of allowed LP modes and the lower the order.

We then draw an **analogy between oxide-confined VCSELs and step-index fibres**. We explain that the VCSEL transverse waveguides are essentially the same as step-index waveguide optical fibres, with oxidized apertures forming the step-index waveguide.

The number of LP transverse modes in a VCSEL emitter is determined by the effective refractive index contrast Δn_{eff} between the circular aperture (“core” region) and region outside of the aperture (“cladding” region). The effective refractive index is defined in Equation (3) in the manuscript. The effective refractive index contrast is determined in Equation (4) in the manuscript. Therefore, the oxide confinement factor Γ_{ox} determines Δn , which in turn determines the allowed number and order of LP modes in the VCSEL. When Γ_{ox} is low, a lower divergence angle is achieved due to a smaller number of allowed LP modes in the VCSEL waveguide. Figure 4a experimentally demonstrates such dependence of the divergence angle on Γ_{ox} .

For conventional extended-cavity VCSELs, due to the long cavity and small FSR, Γ_{ox} cannot be reduced to less than 0.1% while maintaining a single longitudinal mode (or vertical mode, this is different from the lateral/transverse modes discussed above). **Longer cavity length results in smaller vertical mode spacing**. Datacom and consumer electronics VCSELs have no such concern at all because their typical vertical cavity length is only 0.5-2 wavelengths. LiDAR VCSELs, we studied in this article, however, have 10-50 wavelengths. Such long cavities **allow multiple longitudinal modes** within the DBR stopbands. So, there is a limit to how long cavity length we can make without introducing the havoc of multiwavelength lasing. This limit empirically is around 30 wavelengths for typical AlGaAs DBRs. Given this limit, we then introduce an anti-reflective mirror and a light reservoir to **increase the intensity of the cavity instead of its length**. Therefore, we significantly reduce Γ_{ox} by increasing the weight of the region without oxide layers.

2) It is not appropriate to invent a new name for the device, because nothing is conceptually different from VCSELs. The reservoir design of 1D grating is straightforward. Of course, incorporating it into a commercial device with good performance is nontrivial.

Response 1.2 :

The reviewer's recommendation is adopted. We dropped the name “ARCSEL” from the new version and replaced it with “antireflective VCSEL (AR-VCSEL)”. This should help to avoid misunderstandings by making it apparent that it is still a type of VCSEL.

Reviewer #2 (Remarks to the Author):

The authors report on a new variant of a VCSEL with an additional passive cavity separated by a Bragg mirror (called anti-reflective) from the active cavity. They present a theoretical foundation and experimental results. They also give an extensive overview of the state of the art. The results achieved are impressive, in particular those of the 6 and 14 junction devices. The paper is well written and is suitable for a publication in Nature Communications, although I have my doubts whether a new name (ARCSEL) of the VCSEL is justified.

Response 2.0:

We appreciate the reviewer's favourable review and recommendation for publication of our manuscript. As both reviewers mentioned about the name, we dropped the name of "ARCSEL" from the new version and replaced it with "antireflective VCSEL (AR-VCSEL)". This should make it clear that it still belongs to the VCSEL category, preventing any misconceptions.

There are few issues which should be corrected before a publication.

At line 62, the authors specify the divergence angle as half angle. At line 118 they introduce the D86 beam width. In order to calculate M² factor and brightness, the same quantity (i.e. half widths or 86% or 2nd moments) should be used for divergence and width of the beam waist. The authors must specify which quantities have been used to compute M² factor and brightness in Figs. 3-5.

Response 2.1:

We thank the reviewer's comment and we have made the following changes to the manuscript for consistency and clarity purposes.

We changed in Line 117 "The typical full divergence angle in D86" to "The typical full divergence angle ($\Theta=2\theta$) in D86".

We changed in Line 268 "The M² factor is calculated assuming the beam waist radius" to "The M² factor is calculated assuming the beam waist radius (r)".

We changed in Line 277 "The M² factor, which is proportional to the FF angle" to "The M² factor, which is proportional to the FF angle as $M^2 = \pi r \theta / \lambda = \pi r \Theta / (2\lambda)$ where Θ is the full angle in D86".

We added in Line 284 "The brightness and spectral brightness are calculated using Equations (1-2) and $\Delta\Omega = \pi\Theta^2/4$, where Θ is the full angle in D86."

By the way, I do not understand the sentence "defined as the angle at which the D86 beam width expands in the far field, where the D86 beam width is defined as the diameter of the circle that is centred at the centroid of the beam profile and contains 86% of the beam

power”. It is not possible to establish a connection between a point in the beam waist and an angle in the far field.

Response 2.2:

We apologize for the confusion. The term “beam width” is different from the term “beam waist”. The beam width and beam profile defined in D86 are far-field terms rather than near-field terms. We made the following changes to the sentence.

We replaced the sentence “defined as the angle at which the D86 beam width expands in the far field, where the D86 beam width is defined as the diameter of the circle that is centred at the centroid of the beam profile and contains 86% of the beam power” by “defined as the angle at which the D86 beam width in the far field proportionally increases with the distance from the light source, where the D86 beam width in the far field is defined as the diameter of the circle that is centred at the centroid of the beam far field profile and contains 86% of the beam power”.

Although the authors state that the simulations were performed using Ansys Lumerical FDTD, some more details are required. Was it a semivectorial or a vectorial simulation? Which boundary conditions were used? How the static distribution is obtained?

Response 2.3:

We thank the reviewer’s suggestion to add details of the simulations. We added the following in the Methods part of the manuscript.

“Full vectorial simulation was conducted using perfectly matched layer (PML) boundary condition, and final simulation results were obtained with auto shutoff minimum of 1E-5 and auto shutoff maximum of 1E5. Simulations of both with gain and without gain in the active layer were conducted. The simulated normalized $|E(z)|^2$ field intensity profiles with gain and without gain were compared with a difference less than 0.1%, confirming a stable static distribution.”

Line 79: Give a citation for the LIDAR filters specially designed.

Response 2.4:

We found a reference paper that demonstrates how filters composed of various materials exhibit varying temperature drift coefficients.¹ In contrast to the typical filter, which typically exhibits a temperature-induced wavelength shift of less than 0.01 nm/°C, this paper reveals a more substantial shift of 0.0037%/°C in wavelength when the filter substrate’s coefficient of thermal expansion is near zero, equivalent to 0.024 nm/°C at 650 nm as demonstrated or 0.033 nm/°C extrapolated at 905 nm, though still falls short of the 0.07 nm/°C benchmark.

Our sources in the supply chain imply that a filter product with a factor of 0.07 nm/°C matching VCSEL exists and is supplied to some LiDAR companies. However, there is no publication or direct evidence yet. Therefore, to prevent possible misinformation, we have adjusted our wording as follows: “LiDAR filter specially designed to match VCSELs’ 0.06 ~0.07nm/°C is possible.”

Line 83: Better use elliptic instead of asymmetric.

Response 2.5:

We thank the reviewer for the suggestion and have changed the word “asymmetric” to “elliptic” in Line 83 of the manuscript.

Eq. (3): E is a complex-value field because one has to deal with outgoing fields at the surfaces of the devices, so that n_{eff} is complex, too? The same question concerns Eq. (5). What is the meaning of the imaginary part of the confinement factor?

Response 2.6:

We thank the reviewer for pointing out the confusion. In Equation (3) and (5) we meant to use $|E(z)|^2$ field intensity rather than the complex-value field. We have made changes to Equation (3) and (5) and main text accordingly.

Equation (3) has been changed from $n_{eff} = \frac{\int n(z)E^2(z)dz}{\int E^2(z)dz}$ to $n_{eff} = \frac{\int n(z)|E(z)|^2 dz}{\int |E(z)|^2 dz}$

Equation (5) has been changed from $\Gamma_{ox} = \frac{\int_{Oxide} E^2(z)dz}{\int_{Total\ cavity} E^2(z)dz}$ to $\Gamma_{ox} = \frac{\int_{Oxide} |E(z)|^2 dz}{\int_{Total\ cavity} |E(z)|^2 dz}$

Line 167 “ $E^2(z)$ is the electric field intensity in the z-axis direction” has been changed to “ $|E(z)|^2$ is the electric field intensity in the z-axis direction”.

Reviewer #3 (Remarks to the Author):

In this paper, the authors propose and experimentally demonstrate a new device structure of VCSELs to realize single-longitudinal-mode lasing while decreasing their divergence angles. The authors perform systematic numerical analyses and repeated experiments. In addition, the paper is logically organized with thorough descriptions on the history of high-power VCSELs. Therefore, this reviewer believes that the paper is of interest for researchers and engineers concerning the development of high-power VCSELs for LiDAR applications. Having said that, considering the fact that various efforts have been made to increase the brightness of VCSELs in the past, this reviewer is not confident that this work will be of significance to the field of photonics and other related fields. The main concerns with this paper are described below.

Response 3.0:

We express our gratitude to the reviewer for acknowledging the high-quality results presented in this manuscript. Nevertheless, we respectfully differ from the reviewer's assessment regarding the significance of this work. We firmly contend that the findings reported herein are eminently suitable for publication in Nature Communications, and the reasons are as follows:

1). **Antireflective VCSEL** (or AR-VCSEL, renamed as in responses 1.2 and 2.0) has significantly advanced the brightness of VCSELs, **achieving numerous world records** in this article, including **small-angle emission, brightness, spectral brightness, and optical power density in VCSEL arrays**. The revised version further includes **the highest power single-mode VCSEL**. With such a wealth of world records, and arguably the most impactful innovation in this field in the past decade, we believe this work merits publication in a Nature subseries journal.

2). AR-VCSEL **revolutionized the fundamental structure of traditional VCSELs** and represents **a significant breakthrough in VCSEL research over the past decade**. **VCSELs are the most widely used lasers in human history**, numbering in the billions, surpassing all other laser types combined. VCSELs have become an integral part of our daily lives, with nearly every smartphone user having been exposed to the infrared light from VCSELs (used in proximity sensors, face ID, TOF cameras, etc.). This includes most authors, reviewers, editors, and readers of the Nature series. Not to mention that Apple's entire product line incorporates multiple VCSELs, and most Android smartphones use VCSEL-equipped proximity sensors. Our startup, Vertlite, alone has shipped more than 100 million VCSEL chips to date. The VCSEL's vertical structure has remained unchanged for the past two decades due to its maturity, until its recent application in LiDAR. **AR-VCSEL as a disruptive innovation in such most widely used lasers has a profound impact on everyday life**. We believe that readers of the Nature series would be curious to learn about the lasers that surround them daily, the advancements in this technology, and its future applications in automobiles.

3) AR-VCSEL enables > 200 m long range that traditional VCSEL cannot provide. With its low cost, it is changing the game of the industry. With Cruise and Waymo's robotaxi hitting the roads in San Francisco, humanity is officially entering the era of autonomous driving, in which LiDAR is critical. Currently, several of the world's largest LiDAR companies (e.g. HESAI and Ouster) are providing VCSEL-based products. Due to the cost-effectiveness of VCSEL technology, VCSEL-based LiDAR has the potential to bring down the cost of LiDAR systems to less than \$100 in the next 3 ~ 5 years. While conventional long-cavity VCSELs face limitations in terms of divergence angle and long-range LiDAR applications, our antireflective VCSEL technology enables LiDAR to achieve distances exceeding 200 meters, as confirmed by feedback from our LiDAR customers. This breakthrough is poised to reshape the entire LiDAR industry ecosystem, with more mainstream LiDAR companies likely to adopt the AR-VCSEL path.

4) Extended applications: While the innovative work presented in this paper is a direct response to the need for increased brightness in LiDAR applications, it should be noted that this technology extends beyond LiDAR alone. Toward the end of the paper, we mention potential applications in other areas, such as single mode laser for data communications. The concept of light reservoirs has implications for other optical fields, **including PCSELs and TCSELs**. Consequently, we believe that the impact of this technology is substantial, influencing the design principles of all surface-emitting lasers.

In light of the foregoing, we firmly maintain that the findings presented in this manuscript merit publication within the esteemed pages of Nature Communications.

1. As explained by the author, the use of multi-junction active layers or an extended cavity has already been investigated for increasing the output and brightness of VCSELs. In this paper, the authors employ antireflective layers in addition to the above-mentioned structures in order to further reduce the oxide confinement factor and to suppress higher-order modes. However, the concept of reducing the oxide confinement factor itself is the same as that of the extended cavity, and the only difference is the method to embody that concept. Furthermore, although the beam divergence angle is improved to some extent, multiple transverse modes still oscillate in the proposed device. For these reasons, the results presented in this paper can be regarded as incremental progress in this field, and they might be of little interest for researchers in other fields.

Response 3.1.1:

We respectfully disagree that it is just an incremental of the conventional extended cavity approach because:

1). A new structure and a new design methodology of the surface-emitting laser are introduced. Instead of simply increasing the cavity length by the traditional method, we start to engineer the cavity E-field strength by introducing the antireflective light reservoir. Both the structure and the design methods add new tools to the surface-emitting laser

technology, e.g., AR-VCSEL structure has the potential to form quasi-3D photonic crystals and solve the incompatibility of PCSEL/TCSEL with multijunction.

2) Like the 3D gate design of the FinFET transistor has enabled the Si technology node of 14 nm and below, AR-VCSEL has **enabled the LiDAR VCSEL “technology node” of 16° or below, which is a technological watershed for 200m long-range** (which is an industrial consensus for how far the main LiDAR needs to see). Without the E-field enhancement, the traditional approach cannot overcome the divergence barrier of ~ 16° for VCSELs with > 5 junctions. Therefore, its impact is not just “incremental”, but game-changing.

3). Not limited to VCSELs, no other laser technology can compete with AR-VCSELs in commercial LiDAR requiring < 16° divergence under the same cost. Therefore, **AR-VCSEL is irreplaceable in the next technology node of LiDAR**. Our AR-VCSELs are very popular among LiDAR customers. We have seen the trend that AR-VCSEL is gradually replacing other LiDAR light sources such as EEL and traditional extended VCSELs. The impact will be far-reaching.

Given the novel design methodology introduction, and its irreplaceability for the next generation of long-range LiDAR, AR-VCSEL is truly a game-changing advancement in semiconductor laser technology.

It should be also noted that the value of M2 shown in Fig. 3 was calculated assuming an aperture of a single VCSEL, which is actually incorrect and misleading. Considering the array device with a diameter of 250 μm, the theoretical diffraction limit of the device is around 0.3°. Therefore, the value of M2 should be as large as ~30.

Response 3.1.2:

We thank the reviewer for raising the concern and we are pleased to address the confusion.

In a VCSEL array, each emitter's laser beam is independent and incoherent with any other emitter in the array. The M^2 in figure 3 means the average of M^2 of all emitters in the array rather than the M^2 of the whole array. The far field of the array is simply the superposition of the far fields of all individual emitters in the array since the distance between emitters (< 250 μm) is negligible at far distance of ~115 cm (shown in Figure R1 below). Because all emitters are nearly identical, their far field angles are also nearly identical, so their superposition as the whole array.

For clarity purposes, we have modified the description of the M^2 factor in the manuscript as follows:

In the Figure 3 caption, we have changed “Comparison of the calculated M^2 factor” to “Comparison of the averaged M^2 factor of all individual emitters in an array” and have changed “The M^2 factor is calculated” to “The averaged M^2 factor of all individual emitters in an array is calculated”.

In Line 277 we have changed “M² factor” to “averaged M² factor of all individual emitters in an array”.

Figure R1. Schematic illustration of the beam profiles of all emitters in an array. **a**, Device layout. **b**, Beam profiles showing mutually incoherently beams. **c-e**, Schematics of the far field diameter at different distances from the device.

Figure R2. Revised Figure 3 in the manuscript.

2. Relating to the above comment, in order to demonstrate complete suppression of multiple transverse modes, the authors should also carefully adjust the aperture size of each VCSEL. In order to validate the effectiveness of the proposed concept, it is necessary to demonstrate single-transverse-mode lasing in the proposed VCSELs with a larger oxidation aperture than conventional single-mode VCSELs.

Response 3.2:

The primary objective of this paper is to introduce an innovative approach to suppress transverse modes and enhance the brightness of MJ VCSELs, to meet the requirements of

long-range LiDAR applications, achieving a divergence angle of 8-16 degrees, which cannot be achieved through traditional methods. Given the substantial evidence showing the effectiveness of our method, we think whether it achieves single transverse mode lasing should not logically alter the conclusions of this paper. According to our interaction with several major LiDAR manufacturers' R&D engineers, Time-of-Flight LiDAR technology so far does not require the laser to be in single-transverse-mode.

Nevertheless, the reviewer's request remains a valuable suggestion to enhance the strength and completeness of our work. We conducted additional experiments to realize "single-transverse-mode lasing in the proposed VCSELs with a larger oxidation aperture than conventional single-mode VCSELs". The results are shown in the new Figure 5. In addition, we added corresponding texts in our main manuscript and Figures S8 and S9 in the Supplementary Information. All the added parts are as follows.

"In addition to varying the number of junctions and the oxide confinement factor, we investigated another critical parameter: the diameter of the optical aperture (OA). We fabricated a series of AR-VCSEL arrays with a dimension of 250 μm , each densely populated with identical emitters featuring OA diameters ranging from 7 μm to 21 μm , all with the same epitaxial structure as shown in Figure 2. Subsequently, we conducted measurements of their divergence angles and calculated the average M^2 values within these arrays. Our results reveal a clear correlation between OA size and M^2 values as shown in Figure 5a. As the OA size decreases, M^2 values also decrease. Notably, when the aperture size is reduced to 7 μm , the M^2 approaches a value close to 1, suggesting that the majority of emitters might operate in nearly a single transverse mode lasing regime.

To delve deeper into the behavior of the 7 μm OA array sample and confirm the possibility of single-mode lasing, we employed a 100 ns pulse driver for testing. This allowed us to gain better control at lower currents, facilitating the determination of the transition point between single-mode and multimode lasing. We utilized a free-space lens (as depicted in Figure S9) to couple the emission of this specific emitter into an optical fiber. At an optical power of 28.4 mW with a current of approximately 7.8 mA (calculated by dividing the total array current by the number of emitters) and a current density of around 200 A/ mm^2 , the spectrum indicates single-mode lasing with a side mode suppression ratio (SMSR) reaching nearly 40 dB. This confirms the achievement of high-power single transverse mode lasing in AR-VCSELs under the 100 ns pulse condition. The 28.4 mW peak power of this individual emitter surpassed the highest power of 14 mW for MJ VCSEL single-mode lasing reported in the literature⁴⁷. Notably, the 7 μm OA of the AR-VCSEL is considerably larger than the 3 ~ 4 μm typically required for traditional single-mode VCSELs without additional surface relief or complex structures.

Additionally, we obtained both near-field and far-field images of the light emitted from the aperture. The optical fields exhibit slightly elliptical shapes, likely attributed to the imperfect circular shape of the aperture. When analyzing the side views of the far-field

intensity, their shapes closely resemble Gaussian curves, a key characteristic of single-mode operation.

It is essential to acknowledge that, at this stage, we cannot conclusively confirm the consistency of the optical mode between continuous wave (as in most previous single-mode VCSEL work) and pulsed conditions (as in this work). This aspect necessitates further investigation in the future. Furthermore, it's important to note that in AR-VCSEL arrays, as well as in regular VCSEL arrays, the light emitted by individual emitters lacks coherence with each other. Consequently, even if each emitter operates in a single mode, the light field of the entire array cannot be considered a single mode. Instead, it represents a superposition of multiple single-mode light fields."

Figure R3 (The new Figure 5 in the manuscript) Demonstration of single-transverse-mode AR-VCSEL. **a**, The averaged M^2 factor of all individual emitters in an array for AR-VCSELs with optical aperture (OA) sizes varying from 7 μm to 21 μm in diameter at different current densities, with the epi structure the same as in Figure 2. **b**, The measured single emitter lasing spectra of the OA 7 μm AR-VCSEL at different output powers, showing single transverse mode operation at 28.4 mW with a side mode suppression ratio of ~ 40 dB. **c**, The measured near field and far field images of the OA 7 μm AR-VCSEL at

different light output powers from 18.3 mW to 35.4 mW. b-c were measured at 100 ns pulse condition and a repetition rate of 20 kHz.

In this summary, we added “By varying the oxidation aperture sizes, 28.4 mW high-power single transverse mode lasing was realized in a 7 μm diameter 6-junction AR-VCSEL single emitter.”

Figure R4 (The new Figure S8 in the Supplementary Information) Device images of AR-VCSEL arrays having optical aperture sizes ranging from 7 μm to 21 μm in diameter.

Figure R5 (The new Figure S9 in the Supplementary Information). Schematic of the free-space lens setup for single-emitter spectrum measurement.

3. The discussion regarding the comparison between VCSELs and PCSELs is highly biased. The authors consider that higher power density of VCSELs is one important advantage over PCSELs, but the power density is simply determined by the current density injected into the device, and is not inherent to the device structure. For example, by fabricating a PCSEL with a diameter of 100 μm and injecting high-amplitude current into it, a power

density of 1000 W/mm² or more can be easily obtained, with a much narrower divergence angle than VCSELs. In addition, higher power densities are not always desirable because it leads to shorter device lifetimes. Furthermore, as for the discussion of the cost, it is necessary to include the cost of the lens system to shape the array emission into a parallel narrow-divergence beam, because PCSELs enable lens-free operation.

Response 3.3:

We aim to achieve the utmost objectivity and fairness in our comparisons. We adhere to the principle of letting data speak for itself. Our analysis of PCSEL is based on publicly available literature. Just as we acknowledge in the original Figure 5 that AR-VCSEL surpasses traditional VCSEL in brightness, there is still a two-order-of-magnitude gap between them and the best PCSEL. Conversely, in terms of the power density, PCSEL does indeed have a significant gap compared to the best EEL, VCSEL, and AR-VCSEL. Previous research literature on PCSEL has not highlighted its weaknesses in power density. We believe it is necessary, on the Nature series platform, to provide readers with a comprehensive understanding of the strengths and weaknesses of various lasers, including PCSEL, in LiDAR applications. We believe that the current commercialization of PCSEL technology for LiDAR applications faces significant challenges in terms of price competitiveness. However, the challenges in the LiDAR field do not necessarily indicate that PCSEL lacks tremendous potential in other application areas.

Next, we will address the specific technical arguments raised by the reviewer one by one:

Firstly, the reviewer claims that “power density is simply determined by the current density injected into the device”. This is a misunderstanding. The number of photons that can be generated by a single carrier passing through can vary significantly, and this is determined by the External Quantum Efficiency (EQE), which is dictated by the device's structure. In Supplementary Information Figure S10d, we demonstrate that a 14J VCSEL can achieve an EQE of up to 800%. The best EQE achieved with PCSEL so far is 52%³, indicating a fifteen-fold difference. PCSEL currently cannot achieve the same multi-junction design, and it is also challenging in the future (as explained in the penultimate paragraph of the supplementary information). The EQE alone represents an order of magnitude difference, which is “inherent to the device structure”.

Secondly, we have included the highest optical power densities found in the PCSEL literature in original Figure 5. There is nearly a two-order-of-magnitude difference compared to VCSEL. The highest power density achieved with PCSEL is only 60W/mm², while our AR-VCSEL has experimentally reached 5000W/mm². We agree with the reviewer that increasing the injection current further may increase the power density of PCSEL (although beam quality may degrade, and angles may increase, as detailed in the

last paragraph of the supplementary information). However, since we can only reference publicly available articles, we have not come across any published PCSEL data approaching 100 W/mm^2 , let alone the reviewer's mentioned 1000 W/mm^2 . If the reviewer can provide references with experimental data for higher power densities than our existing citations, we would be happy to modify original Figure 5 and related comments accordingly.

Thirdly, the reviewer points out that “higher power densities are not always desirable because it leads to shorter device lifetimes.” What we aim for is to simultaneously achieve both sufficient lifetime and adequate power densities. One of our AR-VCSELs, which is currently in production, has been reliably validated for longevity at $\sim 1000 \text{ W/mm}^2$, exceeding the lifetime requirements for LiDAR applications. If the reviewer is interested, we can provide a complete AECQ report. Another AR-VCSEL we are developing has also met the longevity requirements at near 2000 W/mm^2 . While we cannot confirm when the device with $\sim 5000 \text{ W/mm}^2$ can meet the longevity requirements, we are confident that it could be achieved within a couple of years.

Finally, regarding the cost, the reviewer suggests evaluating the system cost, including the lens. We have indeed acknowledged this point in the manuscript: “PCSEL offers $\sim 0.1^\circ$ and can significantly reduce the size of the collimation lens or possibly eliminate it for shorter distances or lower resolutions. In this sense, PCSEL has advantages with a smaller etendue allowing a smaller spot size after collimation.” However, the cost of lenses used solely for the transmitter accounts for less than 20% of the total system cost. Even if we can eliminate the lens for the laser, the lens for the receiver cannot be eliminated, and in some LiDAR optical paths, the transmitter and receiver can share at least some of the lenses. Because the cost of lasers in LiDAR already accounts for about 20 ~ 40% of the total cost, if the laser area needs to be increased tenfold, the laser alone would cost 2 ~ 4 times the total budget of the whole LiDAR system. Even if we save 20% by eliminating lenses, it wouldn't make a significant difference. Therefore, optical power density remains a crucial factor for low-cost considerations.

In the revised manuscript, we have provided corresponding supplements: “Under current PCSEL technology, even if we completely eliminate the cost of the transmitting-end lens (< 20% of LiDAR total cost), the savings are far less than the tenfold (conservative estimate) increase in total cost brought about by the larger chip area (2 ~ 4 times that of LiDAR's total cost).”

Considering all these concerns, this reviewer cannot conclude that this paper has enough novelty and impact to worth publication in Nature Communications. This reviewer

recommends the authors to resubmit the paper to another specialized journal such as Optics Express.

Response 3.4:

Based on the above responses and revisions, we hope the reviewer is convinced and considers our work suitable for publication in NATURE COMMUNICATIONS.

References

1. Fredell, M., Jr, T. R., Cote, W., Mann, R. & Jr, R. J. Stable and tunable performance of ultra-narrow bandpass and high edge slope dichroic optical filters. in *Free-Space Laser Communications XXXII* vol. 11272 323–330 (SPIE, 2020).
2. Dummer, M. M., Ghods, A., Xu, G., Rothwell, S. & Johnson, K. Single-mode multi-junction VCSELs with integrated transverse mode filter. in *Vertical-Cavity Surface-Emitting Lasers XXVII* vol. 12439 58–65 (SPIE, 2023).
3. Yoshida, M. *et al.* Photonic-crystal lasers with high-quality narrow-divergence symmetric beams and their application to LiDAR. *J. Phys. Photonics* **3**, 022006 (2021).

REVIEWERS' COMMENTS

Reviewer #1 (Remarks to the Author):

The authors have addressed my concerns in a satisfying way. I recommend it for publication.

Reviewer #2 (Remarks to the Author):

The authors partially revised the manuscript. However, as far as I see they only gave a lengthy answer to question 1 of Reviewer 1 in the rebuttal letter but did not modify the manuscript correspondingly. The authors must catch up on this.

Reviewer #3 (Remarks to the Author):

In this revision, the authors have addressed the concerns raised in the first round of the review. Especially, the results of single-transverse-mode high-power VCSELs based on the concept of anti-reflective layers are important, by which the paper is more strengthened than the original one. Therefore, this reviewer considers that the paper is now suitable for publication in Nature Communications.

Response Letter to Reviewer Comments

(Black for comments; green for responses; blue for revisions)

Reviewer #1 (Remarks to the Author):

The authors have addressed my concerns in a satisfying way. I recommend it for publication.

Response 1.0:

We appreciate the reviewer's recommendation for publication.

Reviewer #2 (Remarks to the Author):

The authors partially revised the manuscript. However, as far as I see they only gave a lengthy answer to question 1 of Reviewer 1 in the rebuttal letter but did not modify the manuscript correspondingly.

The authors must catch up on this.

Response 2.0:

We appreciate the reviewer's comment. We have made the following modifications to the manuscript correspondingly.

To facilitate the understanding of the reason for reduction of divergence angle, we added "The beam quality of each LP mode depends on the mode order. The lowest order mode, or the so-called fundamental mode, has the highest beam quality, or the lowest M^2 factor ($M^2 = 1$), and therefore the smallest divergence angle once the mode is coupled from the waveguide into free space. The higher the mode order, the higher the M^2 factor ($M^2 > 1$) and the larger the divergence angle." after the first sentence in the paragraph after Equation (5).

Reviewer #3 (Remarks to the Author):

In this revision, the authors have addressed the concerns raised in the first round of the review. Especially, the results of single-transverse-mode high-power VCSELs based on the concept of anti-reflective layers are important, by which the paper is more strengthened than the original one. Therefore, this reviewer considers that the paper is now suitable for publication in Nature Communications.

Response 3.0:

We appreciate the reviewer's recognition on the significance of the single transverse mode work and recommendation for publication.